# Algoritmos Genéticos para la Fusión Eficiente de Redes Bayesianas mediante Eliminación de Enlaces Previa a la Fusión

**Pablo Torrijos**
Departamento de Sistemas Informáticos
Universidad de Castilla-La Mancha
Albacete, España
Pablo.Torrijos@uclm.es

**José A. Gámez**
Departamento de Sistemas Informáticos
Universidad de Castilla-La Mancha
Albacete, España
Jose.Gamez@uclm.es

**José M. Puerta**
Departamento de Sistemas Informáticos
Universidad de Castilla-La Mancha
Albacete, España
Jose.Puerta@uclm.es

**Juan Á. Aledo**
Departamento de Matemáticas
Universidad de Castilla-La Mancha
Albacete, España
JuanAngel.Aledo@uclm.es

## Abstract

La fusión de Redes Bayesianas (RBs) combina redes de entrada en una sola estructura, equilibrando la conservación de dependencias con la viabilidad computacional. Mientras que una fusión sin restricciones mantiene todas las dependencias, tiende a generar redes excesivamente complejas con un alto *treewidth*, lo que afecta a la escalabilidad de la inferencia. La fusión limitada mitiga este problema al eliminar enlaces para controlar el *treewidth*, aunque puede sobreajustarse al ruido de las redes de entrada y omitir dependencias importantes en las RBs originales. Este artículo presenta un marco de consenso que prioriza las estructuras compartidas entre las redes de entrada mientras impone restricciones al *treewidth*, asegurando un buen consenso. Introducimos algoritmos genéticos con inicialización avanzada, operadores especializados y una función de *fitness* adaptada. Además, adaptamos métodos existentes a este problema e implementamos algoritmos voraces para su evaluación comparativa. Los experimentos realizados con RBs sintéticas y reales demuestran la superioridad de los algoritmos genéticos propuestos frente a los métodos adaptados y los voraces usados como base.

## 1 Introducción

Las Redes Bayesianas (RBs) [6, 8] son modelos gráficos probabilísticos que representan dependencias complejas entre variables mediante Grafos Dirigidos Acíclicos (*DAGs*, del inglés *Directed Acyclic Graph*), combinando una estructura gráfica que codifica dependencias condicionales con distribuciones de probabilidad que cuantifican estas dependencias. Son interpretables gracias a su estructura gráfica y permiten inferencia simbólica [6], lo que las hace útiles en áreas como bioinformática [1], salud [11] o modelado ambiental [5], donde permiten tomar decisiones bajo incertidumbre. Además, su capacidad para representar relaciones causales y facilitar la interpretación de resultados resalta su importancia en el ámbito de la Inteligencia Artificial (IA) explicable [2].

El aprendizaje de RBs a partir de datos, aunque NP-duro [4], ha sido ampliamente investigado [17, 10]. La fusión estructural de RBs [13, 14] es fundamental para combinar redes provenientes de diversos expertos o fuentes de datos. Esto resulta especialmente relevante en el aprendizaje

XVI Congreso Español de Metaheurísticas, Algoritmos Evolutivos y Bioinspirados (MAEB 2025).

distribuido y federado, donde fusionar RBs es esencial para agregar modelos locales, como se muestra en trabajos recientes [9, 16]. Lograr una fusión óptima implica construir un modelo que capture todas las dependencias condicionales de las redes de entrada. Sin embargo, la fusión estándar de RBs genera grafos densos y complejos, limitando su uso en escenarios que requieren inferencia eficiente e interpretabilidad. Aunque este problema es NP-duro [13], [14] propuso un enfoque heurístico para aproximar la RB fusionada óptima.

La complejidad de la inferencia en una RB depende de su *treewidth* $t$, que se define como el tamaño del subconjunto más grande de nodos completamente conectados en una versión triangulada del grafo moral de la red[1]. A medida que $t$ aumenta, la complejidad crece exponencialmente, $O(n \cdot k^{t+1})$ [3], donde $n$ es el número de variables y $k$ el número de estados por variable, lo que hace que las RBs con alto *treewidth* sean inviables para su uso práctico. Para mitigar esto, [15] propuso un algoritmo genético que elimina enlaces en la red fusionada, reduciendo su *treewidth* mientras conserva la estructura esencial de la fusión sin restricciones.

La computación evolutiva se ha utilizado históricamente como solución heurística en diversos problemas relacionados con las RBs [10]. En este trabajo, proponemos una metodología de fusión para lograr el *treewidth* deseado en la RB de consenso mediante la eliminación selectiva de enlaces en las RBs originales antes de la fusión, utilizando un algoritmo genético con operadores ajustados al problema. Diseñamos dos enfoques: uno donde cada aparición de un enlace se trata como un elemento único en el cromosoma y otro donde los enlaces idénticos se tratan como un solo elemento. Para cada estrategia, se desarrolla un algoritmo voraz como base para la comparación de rendimiento.

Nuestra propuesta difiere de [15] al redefinir la métrica de calidad de la solución. Mientras que la fusión en [14] captura todas las independencias condicionales, en algunos casos no es deseable preservar completamente esta fidelidad, especialmente en escenarios como el aprendizaje federado, donde independencias erróneas o adversariales pueden corromper la RB fusionada. Así, minimizamos la divergencia entre la RB fusionada y las de entrada, mejorando la robustez frente a (in)dependencias inexactas o maliciosas.

Las principales contribuciones de este trabajo son:

- Establecer una nueva definición de consenso para la fusión de RBs que minimiza la divergencia promedio entre la RB fusionada y las de entrada mientras cumple con un *treewidth*.

- Proponer un algoritmo genético y uno voraz con dos codificaciones diferentes para lograr el consenso de RBs mediante la eliminación selectiva de enlaces en las RBs de entrada.

- Introducir nuevas métricas para evaluar la calidad de las RBs fusionadas, enfocándose en la divergencia respecto a las redes de entrada.

- Proveer el código para permitir la reproducibilidad y fomentar la investigación en el campo.

El resto del artículo está estructurado de la siguiente manera: la Sección 2 presenta una visión general de las RBs y conceptos clave. La Sección 3 formaliza el problema de consenso y sus restricciones. La Sección 4 describe los algoritmos genéticos y voraces propuestos para lograr el consenso bajo una restricción de *treewidth*. La Sección 5 muestra los resultados de experimentos con datos sintéticos y reales. Finalmente, la Sección 6 resume los hallazgos y señala futuras líneas de investigación.

## 2 Preliminares

### 2.1 Redes Bayesianas

Una Red Bayesiana (RB) [6, 8] es un modelo gráfico probabilístico que representa las dependencias (e independencias) condicionales entre un conjunto de variables $\mathcal{X} = \{X_1, \ldots, X_n\}$. Formalmente, una RB se define como un par $\mathcal{B} = (\mathcal{G}, \mathcal{P})$, donde $\mathcal{G} = (\mathcal{X}, \mathcal{E})$ es un Grafo Acíclico Dirigido (DAG) cuyos nodos $\mathcal{X}$ representan las variables y los bordes dirigidos $\mathcal{E}$ codifican las dependencias entre estas variables, y $\mathcal{P}$ es un conjunto de distribuciones de probabilidad condicionales, $\mathbb{P}(X_i \mid pa(X_i))_{i=1}^{n}$, donde $pa(X_i)$ denota el conjunto de padres de $X_i$ en $\mathcal{G}$.

---

[1]El grafo moral de una RB es el grafo no dirigido obtenido al vincular todos los pares de nodos no adyacentes con un hijo común y luego eliminar las direcciones de las enlaces.

La estructura $\mathcal{G}$ codifica las independencias condicionales utilizando el criterio de *d-separación* [8], lo que permite la factorización de la distribución de probabilidad conjunta $\mathbb{P}(X_1, \ldots, X_n)$ como $\mathbb{P}(X_1, \ldots, X_n) = \prod_{i=1}^{n} \mathbb{P}(X_i \mid pa(X_i))$. Así, $\mathcal{G}$ impone que cada variable $X_i$ es condicionalmente independiente de sus no-descendientes dado sus padres $pa(X_i)$ en el DAG.

Las independencias codificadas por $\mathcal{G}$ se representan como $I(\mathcal{G})$, donde cada elemento corresponde a una relación de independencia condicional de la forma $(X_i \perp X_j \mid Z)$, con $X_i$ y $X_j$ siendo condicionalmente independientes dado un subconjunto de nodos $Z$. Un DAG $\mathcal{G}_1$ es un *I-map* (Mapa de Independencias) de otro DAG $\mathcal{G}_2$ si $I(\mathcal{G}_1) \subseteq I(\mathcal{G}_2)$, es decir, si $\mathcal{G}_1$ codifica al menos las mismas independencias que $\mathcal{G}_2$. Un DAG $\mathcal{G}_1$ es un *I-map mínimo* de $\mathcal{G}_2$ si no se puede eliminar ningún arco de $\mathcal{G}_1$ sin violar al menos una independencia condicional en $\mathcal{G}_2$. Así, un *I-map mínimo* $\mathcal{G}_1$ es el grafo más reducido que retiene todas las independencias condicionales de $\mathcal{G}_2$.

## 2.2 Fusión estructural de RBs

La fusión estructural de Redes Bayesianas combina varias RBs, provenientes de diferentes fuentes o expertos, en una sola RB que capture las dependencias esenciales. Sea $\mathbb{B} = \{\mathcal{B}_1, \ldots, \mathcal{B}_r\}$ un conjunto de RBs con los DAGs correspondientes $\mathbb{G} = \{\mathcal{G}_1, \ldots, \mathcal{G}_r\}$. El objetivo es construir un DAG $\mathcal{G}^+$ que sea un *I-map* mínimo de la intersección de los conjuntos de independencia, es decir, $I(\mathcal{G}^+) = \bigcap_{i=1}^{r} I(\mathcal{G}_i)$, minimizando el número de arcos y capturando las independencias comunes.

Por ejemplo, con las tres RBs representadas por los DAGs en la Fig. 1, el objetivo es obtener un DAG $\mathcal{G}^+$ que mantenga solo las independencias condicionales comunes, reduciendo el número de arcos. La Fig. 3 ilustra dos posibles fusiones, donde la estructura de la izquierda es óptima. La construcción

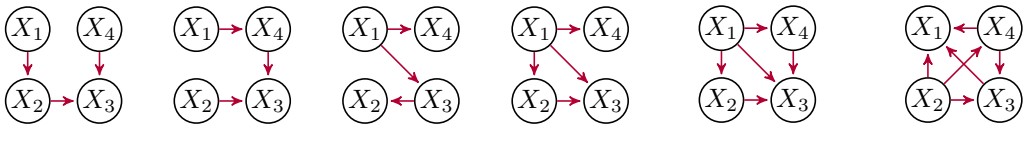

Figura 1: $\mathcal{G}_1 = \mathcal{G}_1^{\sigma^*}$; $\mathcal{G}_2 = \mathcal{G}_2^{\sigma^*}$; $\mathcal{G}_3$.  Figura 2: $\mathcal{G}_3^{\sigma^*}$.  Figura 3: $\mathcal{G}^{+\sigma^*}$; $\mathcal{G}^{+(X_2, X_4, X_3, X_1)}$.

de $\mathcal{G}^+$ se realiza mediante los siguientes pasos [13]: (1) Definir un orden $\sigma$ sobre las variables $\mathcal{X}$. (2) Modificar cada DAG $\mathcal{G}_i$ para obtener un *I-map* mínimo $\mathcal{G}_i^{\sigma}$ de $\mathcal{G}_i$ consistente[2] con $\sigma$. (3) Construir $\mathcal{G}^+$ como la unión de los arcos de todos los $\mathcal{G}_i^{\sigma}$, es decir, $\mathcal{G}^+ = \bigcup_{i=1}^{r} arcs(\mathcal{G}_i^{\sigma})$, donde $arcs(\mathcal{G})$ devuelve el conjunto de arcos dirigidos en $\mathcal{G}$.

El *Método A* [13] se aplica en el paso (2), manteniendo las independencias condicionales de cada $\mathcal{G}i$ codificadas por $d$-separación. Esto garantiza que $\mathcal{G}^+$ sea un *I-map* mínimo de la intersección de los conjuntos de independencia de todos los grafos de entrada, es decir, $I(\mathcal{G}^+) = \bigcap_{i=1}^{r} I(\mathcal{G}_i)$. Para los fundamentos teóricos del *Método A*, se remite al lector a [13].

La elección del orden $\sigma$ es clave para determinar la estructura de $\mathcal{G}^+$. El *Método A* aplica este orden a cada DAG de entrada $\mathcal{G}_i$, obteniendo el *I-map* mínimo $\mathcal{G}_i^{\sigma}$ consistente con $\sigma$. Por ejemplo, las Figs. 1 y 2 muestran la transformación de los DAGs de la Fig. 1 utilizando el orden óptimo $\sigma^* = (X_1, X_2, X_4, X_3)$. Este orden $\sigma^*$ determina la RB fusionada mostrada en la Fig. 3 (izda.).

Aunque encontrar el orden óptimo $\sigma$ es NP-duro, las heurísticas existentes [14] ofrecen de forma eficiente soluciones aproximadas que generan redes casi óptimas. Esto es crucial, ya que un $\sigma$ mal elegido puede resultar en fusiones muy complejas. El método de fusión propuesto en [14] captura todas las dependencias condicionales entre los DAGs de entrada $\mathbb{G} = \{\mathcal{G}_1, \ldots, \mathcal{G}_r\}$, pero impone un criterio de "todo o nada": si una dependencia existe en alguna red $\mathcal{G}_i$, debe aparecer en la red fusionada $\mathcal{G}^+$. Este enfoque genera redes densas con un *treewidth* alto, lo que incrementa la carga computacional. Para mitigar esto, proponemos un enfoque de fusión más flexible, logrando un consenso en el subconjunto representativo de dependencias capturado y reduciendo el *treewidth*.

---

[2]Un DAG $\mathcal{G}$ sobre $\mathcal{X}$ se considera consistente con un orden $\sigma$ de $\mathcal{X}$ si, para cada nodo $X \in \mathcal{X}$, todos los padres de $X$ en $\mathcal{G}$ preceden a $X$ en $\sigma$.

### 2.3 Métricas para medir la similitud estructural de las RBs

Existen varias métricas para evaluar la similitud estructural de las RBs. La distancia estructural de Hamming (SHD) [17] cuantifica las diferencias estructurales mediante el número mínimo de modificaciones (adiciones, eliminaciones o inversiones) de arcos necesarias para alinear dos estructuras de RBs, pero no considera las independencias condicionales codificadas en las redes, lo que puede llevar a una representación incorrecta de la similitud [14].

Para abordar esto, podemos utilizar la distancia estructural moral de Hamming (SMHD) [7, 9, 15], que compara los grafos morales de dos RBs; y la *Fusion Simmilarity* (FSIM, introducida en [14], Sección 6.2) que evalúa la similitud estructural basándose en los *I-maps* mínimos de las redes bajo un orden topológico compartido $\sigma$. Mientras que la SMHD se enfoca en las diferencias estructurales, FSIM se centra en medir las independencias condicionales codificadas por las RBs.

## 3 Definición del problema

El método de fusión propuesto en [14] genera una red fusionada $\mathcal{G}^+$ que conserva todas las dependencias de las redes de entrada $\mathbb{G} = \{\mathcal{G}_1, \ldots, \mathcal{G}_r\}$. Aunque es óptimo en la preservación de todas las dependencias condicionales, este enfoque suele dar lugar a una red densa con alto *treewidth*. Para resolver esto, [15] introdujo la *Fusión Estructural Restringida de RBs*, que elimina selectivamente arcos en $\mathcal{G}^+$ para reducir el *treewidth* bajo un límite, minimizando al mismo tiempo la SMHD con respecto a $\mathcal{G}^+$. Este enfoque busca aproximar $\mathcal{G}^+$ a una red más manejable denotada como $\mathcal{G}^+_{tw \le t}$.

Sin embargo, usar $\mathcal{G}^+$ como referencia tiene limitaciones, ya que puede introducir complejidad innecesaria, especialmente si las redes de entrada contienen dependencias ruidosas o irrelevantes. En el ejemplo de la Fig. 4 (con métricas[3] en la Tabla 1), teniendo en cuenta solo las redes $\mathcal{G}$ y no las $\mathcal{H}$, tres redes de entrada (con *treewidth* 2) se fusionan (Figs. 4a, 4b y 4c). La fusión sin restricciones $\mathcal{G}^+$ (Fig. 4d) tiene *treewidth* 5, mientras que la fusión con *treewidth* restringido, $\mathcal{G}^+_{tw \le 3}$ (Fig. 4e), consigue una red con *treewidth* 3 eliminando solamente dos arcos de $\mathcal{G}^+$.

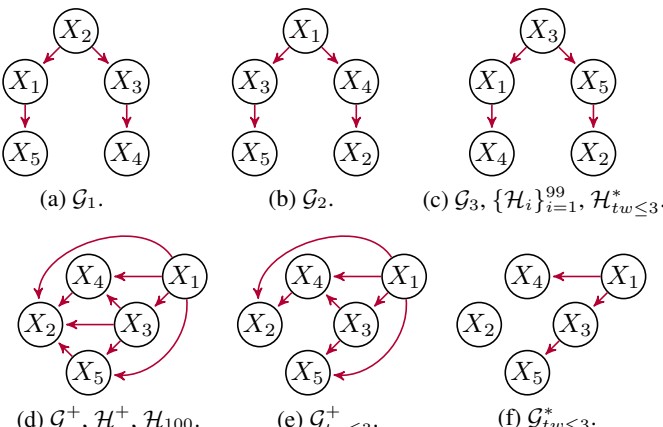

(a) $\mathcal{G}_1$.     (b) $\mathcal{G}_2$.     (c) $\mathcal{G}_3$, $\{\mathcal{H}_i\}_{i=1}^{99}$, $\mathcal{H}^*_{tw \le 3}$.

(d) $\mathcal{G}^+$, $\mathcal{H}^+$, $\mathcal{H}_{100}$.     (e) $\mathcal{G}^+_{tw \le 3}$.     (f) $\mathcal{G}^*_{tw \le 3}$.

Figura 4: Distintas fusiones de $\{\mathcal{G}_1, \mathcal{G}_2, \mathcal{G}_3\}$ y $\{\mathcal{H}_i\}_{i=1}^{100}$.

En la misma Fig. 4, si observamos el ejemplo formado por las RBs $\mathcal{H}$, el efecto del ruido es más pronunciado. Aquí, 99 redes con *treewidth* 2 comparten dependencias (Fig. 4c), pero una red $\mathcal{H}_{100}$ (Fig. 4d) introduce dependencias espurias. Esto puede ocurrir en un escenario de aprendizaje federado con un cliente malicioso. Como resultado, la fusión sin restricciones $\mathcal{H}^+$ (Fig. 4d) replica las dependencias ruidosas de $\mathcal{H}_{100}$. Incluso con el *treewidth* limitado, $\mathcal{H}^+_{tw \le 3}$ (Fig. 4e) no refleja el consenso mayoritario debido a la influencia de $\mathcal{H}_{100}$.

---

[3]TW indica el *treewidth* de la RB; $\text{SMHD}_{\mathcal{G}^+}$ es la SMHD con respecto a la fusión $\mathcal{G}^+$; $\text{SMHD}_{\mathbb{G}}$ es la SMHD media sobre las RBs iniciales $\mathbb{G}$; y $\text{FSIM}_{\mathbb{G}}$ es la FSIM media sobre $\mathbb{G}$

Tabla 1: Métricas para las RBs de la Fig. 4. Los guiones indican casos en los que la métrica no tiene sentido, como cuando implicaría autocomparaciones.

| RB | EJEMPLO 1 | | | | RB | EJEMPLO 2 | | | |
|---|---|---|---|---|---|---|---|---|---|
| | TW | $\text{SMHD}_{\mathcal{G}^+}$ | $\text{SMHD}_{\mathbb{G}}$ | $\text{FSIM}_{\mathbb{G}}$ | | TW | $\text{SMHD}_{\mathcal{H}^+}$ | $\text{SMHD}_{\mathbb{H}}$ | $\text{FSIM}_{\mathbb{H}}$ |
| $\{\mathcal{G}_1,\mathcal{G}_2,\mathcal{G}_3\}$ | 2 | 6 | — | — | $\{\mathcal{H}_i\}_{i=1}^{99}$ | 2 | 6 | — | — |
| | | | | | $\mathcal{H}_{100}$ | 5 | 0 | — | — |
| $\mathcal{G}^+$ | 5 | — | 6 | 7.33 | $\mathcal{H}^+$ | 5 | — | 5.94 | 4.95 |
| $\mathcal{G}^+_{tw\leq3}$ | 3 | **3** | 4.33 | 5.33 | $\mathcal{H}^+_{tw\leq3}$ | 3 | **3** | 3 | 2.99 |
| $\mathcal{G}^*_{tw\leq3}$ | 2 | 7 | **3** | **3.33** | $\mathcal{H}^*_{tw\leq3}$ | 2 | 6 | **0.06** | **0.05** |

Estos ejemplos demuestran cómo usar $\mathcal{G}^+$ como referencia puede llevar a una red fusionada compleja que no refleja adecuadamente las (in)dependencias de las redes de entrada. Proponemos una nueva definición que evita esto y mide la similitud con cada red de entrada $\mathcal{G}_i$. La red fusionada se define como aquella que minimiza la distancia estructural promedio con las redes de entrada bajo una restricción de *treewidth*, capturando las dependencias compartidas y mitigando el ruido o los errores.

**Definición 1 (Consenso estructural restringido de Redes Bayesianas)** *Sea* $\mathbb{G} = \{\mathcal{G}_1, \ldots, \mathcal{G}_r\}$ *el conjunto de DAGs de* $r$ *RBs definidas sobre el mismo dominio. El objetivo del consenso estructural restringido con respecto a los grafos de entrada es obtener un grafo fusionado* $\mathcal{G}^*_{tw\leq t}$, *que maximice la similitud media entre el grafo fusionado* $\mathcal{G}$ *y los grafos de entrada* $\mathcal{G}_i$, *sujeto a una restricción de treewidth, y minimizando una métrica de distancia estructural definida. Específicamente, para un valor máximo de treewidth* $t \in \mathbb{N}$, *con* $t \geq 2$, *la fusión estructural restringida se define como*

$$\mathcal{G}^*_{tw\leq t} = \arg \min_{\mathcal{G}\,;\,tw(\mathcal{G})\leq t} \frac{1}{r}\sum_{i=1}^{r} Metric\left(\mathcal{G}, \mathcal{G}_i\right) \tag{1}$$

*donde* $Metric\left(\mathcal{G}, \mathcal{G}_i\right)$ *representa una métrica de distancia estructural, como* SMHD *o* FSIM, *utilizada para evaluar la similitud entre el grafo fusionado* $\mathcal{G}$ *y cada una de las redes originales* $\mathcal{G}_i$.

Esta definición relaja la noción de *fusión* a un *consenso*, donde la red resultante $\mathcal{G}^*_{tw\leq t}$ captura las dependencias esenciales compartidas entre las redes de entrada, bajo la restricción de *treewidth*. Volviendo al ejemplo de $\mathcal{G}$, en la Fig. 4, aunque las redes de entrada tienen estructuras simples, sus independencias condicionales varían, lo que hace que la fusión sin restricciones $\mathcal{G}^+$ sea compleja. Nuestro enfoque de consenso (Fig. 4f) genera una red $\mathcal{G}^*_{tw\leq3}$ con menos arcos incluso que las redes de entrada, reflejando la falta de consenso en dichas redes iniciales. Las métricas en la Tabla 1 muestran que $\mathcal{G}^+_{tw\leq3}$ logra un mejor $\text{SMHD}_{\mathcal{G}^+}$, mientras que $\mathcal{G}^*_{tw\leq3}$ mejora el $\text{SMHD}_{\mathbb{G}}$ y $\text{FSIM}_{\mathbb{G}}$.

Este efecto es aún más claro en el ejemplo de $\mathcal{H}$ (Fig. 4). El enfoque de consenso $\mathcal{H}^*_{tw\leq3}$ refleja la estructura compartida de $\{\mathcal{H}_i\}_{i=1}^{99}$ y omite el ruido introducido por $\mathcal{H}_{100}$, al contrario que $\mathcal{H}^+$ y $\mathcal{H}^+_{tw\leq3}$. Como se muestra en la Tabla 1, el nuevo enfoque ofrece mejores valores en $\text{SMHD}_{\mathbb{G}}$ y $\text{FSIM}_{\mathbb{G}}$ al replicar la estructura ideal compartida por la mayoría de las redes de entrada.

Una ventaja adicional de este enfoque de consenso es que, a diferencia del método de fusión restringida de [15], que tiende a explotar el *treewidth* máximo permitido $t$ para minimizar $\text{SMHD}_{\mathcal{G}^+}$, genera RBs de consenso $\mathcal{G}^*_{tw\leq t}$ que tienden a mantienen *treewidths* similares o incluso menores que los de las redes de entrada, lo que facilita el aprendizaje y la inferencia eficientes.

## 4 Algoritmo genético para el consenso de RBs bajo restricciones de *treewidth*

Este trabajo presenta un algoritmo genético para el consenso de RBs introducido en la Definición 1 bajo una restricción de *treewidth* ($t$). El enfoque elimina arcos de las redes de entrada para construir una estructura de consenso equilibrando similitud estructural y viabilidad computacional. Mediante la integración de una inicialización voraz y operadores genéticos específicamente diseñados para el problema, el algoritmo garantiza una exploración y explotación eficientes del espacio de búsqueda, respetando la limitación de *treewidth*.

### 4.1 Representación del cromosoma

La representación de un cromosoma $C$ depende de la elección del conjunto de arcos $\mathcal{E}$, que corresponde a los enlaces considerados en el proceso de fusión. Se definen dos configuraciones posibles:

**Sin Repetición ($\mathcal{E}_{\mathbb{G}}$):** $C$ representa todos los arcos únicos de las redes de entrada, es decir, $\mathcal{E}_{\mathbb{G}} = \bigcup_{i=1}^{r}$ enlaces$(\mathcal{G}_i)$. Sea $s = |\mathcal{E}_{\mathbb{G}}|$, con los arcos ordenados lexicográficamente por sus subíndices (por ejemplo, $a_{1,3} \prec a_{1,6} \prec a_{3,4}$). Un cromosoma $C \in \{0,1\}^s$ codifica si cada arco está incluido (1) o excluido (0) en todas las redes de entrada.

**Con Repetición ($\mathcal{E}_{\mathbb{G}}^*$):** $C$ representa los arcos de cada red de entrada por separado, es decir, $\mathcal{E}_{\mathbb{G}}^* = \bigsqcup_{i=1}^{r}$ enlaces$(\mathcal{G}_i)$. Sea $s^* = |\mathcal{E}_{\mathbb{G}}^*|$, con los arcos ordenados lexicográficamente por sus subíndices y la red a la que pertenecen (por ejemplo, $a_{2,3}^1 \prec a_{3,4}^1 \prec a_{1,2}^3 \prec a_{2,3}^3$). Un cromosoma $C \in \{0,1\}^{s^*}$ especifica la inclusión (1) o exclusión (0) de cada arco independientemente en cada red.

**Equilibrio:** La elección entre $\mathcal{E}_{\mathbb{G}}$ y $\mathcal{E}_{\mathbb{G}}^*$ balancea simplicidad y flexibilidad. $\mathcal{E}_{\mathbb{G}}$ define un espacio de búsqueda más pequeño y rápido pero menos específico, mientras $\mathcal{E}_{\mathbb{G}}^*$ permite eliminar enlaces para cada red, pudiendo lograr mejores soluciones a costa de un mayor espacio de búsqueda.

### 4.2 Función de *fitness*

La función de *fitness* evalúa un cromosoma $C$ en función de la similitud del grafo fusionado $\mathcal{G}_C^*$ con las redes de entrada originales $\{\mathcal{G}_1, \ldots, \mathcal{G}_r\}$, penalizando violaciones de la restricción de *treewidth*.

Dado $C$, los grafos de entrada modificados $\{\mathcal{G}_C^1, \ldots, \mathcal{G}_C^r\}$ se definen como:

$$\mathcal{G}_C^i = (\mathcal{X}, \mathcal{E}_C^i); \ \mathcal{E}_C^i = \begin{cases} \{a_{jk}^i \in \text{enlaces}(\mathcal{G}_i) \mid C(a_{jk}^i) = 1\}, & \text{si } \mathcal{E} = \mathcal{E}_{\mathbb{G}}^* \\ \{a_{jk} \in \text{enlaces}(\mathcal{G}_i) \mid C(a_{jk}) = 1\}, & \text{si } \mathcal{E} = \mathcal{E}_{\mathbb{G}} \end{cases}$$

El grafo fusionado $\mathcal{G}_C^*$ se obtiene aplicando el método de la Sección 2.2 [14] a $\{\mathcal{G}_C^1, \ldots, \mathcal{G}_C^r\}$. El *fitness* se calcula como:

$$f(C) = \frac{1}{r} \sum_{i=1}^{r} \text{Metric}(\mathcal{G}_C^*, \mathcal{G}_i) \cdot \begin{cases} 1, & \text{si } \text{tw}(\mathcal{G}_C^*) \leq t, \\ \frac{\text{tw}(\mathcal{G}_C^*)}{t} & \text{e.o.c.}, \end{cases}$$

donde $\text{tw}(\mathcal{G})$ denota el *treewidth* de $\mathcal{G}$. Minimizar $f(C)$ asegura una alta similitud estructural con las redes de entrada, manteniendo la restricción de *treewidth*. El término de penalización guía la búsqueda fuera de soluciones no viables.

### 4.3 Estructura del algoritmo

El algoritmo genético propuesto, detallado en el Alg. 1, resuelve el consenso de RBs bajo una restricción de *treewidth* (Definición 1). La decodificación de cromosomas y el cálculo de la función de *fitness* son tareas computacionales exigentes, por lo que es necesario equilibrar calidad y eficiencia.

---

**Algoritmo 1** Poda de enlaces genética pre-fusión para consenso de RBs

**Requiere:** $\{\mathcal{G}_1, \ldots, \mathcal{G}_r\}$ sobre $\mathcal{X}; t \geq 2; nIts; tamPob; rep$ (bool)
**Asegura:** $\{\mathcal{G}_1^*, \ldots, \mathcal{G}_r^*\}, \mathcal{G}_{tw \leq t}^*$

1: $\mathcal{E} \leftarrow \begin{cases} \bigsqcup_{i=1}^{r} \text{enlaces}(\mathcal{G}_i) & \text{si } rep \\ \bigcup_{i=1}^{r} \text{enlaces}(\mathcal{G}_i) & \text{e.o.c.} \end{cases}$
2: $\begin{cases} \#(a_{jk}^i) \leftarrow \sum_{i=1}^{r} \mathbb{1}(a_{jk}^i \in \mathcal{G}_i), \forall a_{jk}^i \in \mathcal{E} & \text{si } rep \\ \#(a_{jk}) \leftarrow \sum_{i=1}^{r} \mathbb{1}(a_{jk} \in \mathcal{G}_i), \forall a_{jk} \in \mathcal{E} & \text{e.o.c.} \end{cases}$
3: $\{\mathcal{G}_1^*, \ldots, \mathcal{G}_r^*\} \leftarrow$ grafos vacíos definidos sobre $\mathcal{X}$
4: $C^*, C^l \leftarrow \emptyset$      ▷ Mejor cromosoma hasta el momento/en la última it.
5: $\mathcal{C}_{gr} \leftarrow \{\text{Alg. 2}(\{\mathcal{G}_1, \ldots, \mathcal{G}_r\}, t, rep)\}$
6: $\mathcal{C}_{gr} \leftarrow \mathcal{C}_{gr} \cup \{\text{Alg. 2}(\{\mathcal{G}_1, \ldots, \mathcal{G}_r\}, t-1, rep)\}$ si $t > 2$
7: $\mathcal{C} \leftarrow$ inicialización$(tamPob - |\mathcal{C}_{gr}|, \#(a_{jk}), \mathcal{E}) \cup \mathcal{C}_{gr}$
8: **for** $i \leftarrow 1$ to $nIts$ **do**
9:     $C^*, C^l, \{\mathcal{G}_1^*, \ldots, \mathcal{G}_r^*\}, \mathcal{G}_{tw \leq t}^* \leftarrow$ evaluar$(\mathcal{C}, \{\mathcal{G}_1, \ldots, \mathcal{G}_r\})$
10:     $\mathcal{C} \leftarrow$ selección$(\mathcal{C})$
11:     $\mathcal{C} \leftarrow$ cruce$(\mathcal{C})$
12:     $\mathcal{C} \leftarrow$ mutación$(\mathcal{C})$
13:     $\mathcal{C} \leftarrow \mathcal{C} \cup \{C^*, C^l\}$      ▷ Elitismo
14: **end for**
15: Return $\{\mathcal{G}_1^*, \ldots, \mathcal{G}_r^*\}, \mathcal{G}_{tw \leq t}^*$

---

**Algoritmo 2** Poda de enlaces voraz pre-fusión para consenso de RBs

**Requiere:** $\{\mathcal{G}_1, \ldots, \mathcal{G}_r\}$ sobre $\mathcal{X}; t \geq 2; rep$ (bool)
**Asegura:** $\{\mathcal{G}_1^*, \ldots, \mathcal{G}_r^*\}, \mathcal{G}_{tw \leq t}^*$

1: $\mathcal{E} \leftarrow \begin{cases} \bigsqcup_{i=1}^{r} \text{enlaces}(\mathcal{G}_i) & \text{si } rep \\ \bigcup_{i=1}^{r} \text{enlaces}(\mathcal{G}_i) & \text{e.o.c.} \end{cases}$
2: $\begin{cases} \#(a_{jk}^i) \leftarrow \sum_{i=1}^{r} \mathbb{1}(a_{jk}^i \in \mathcal{G}_i), \forall a_{jk}^i \in \mathcal{E} & \text{si } rep \\ \#(a_{jk}) \leftarrow \sum_{i=1}^{r} \mathbb{1}(a_{jk} \in \mathcal{G}_i), \forall a_{jk} \in \mathcal{E} & \text{e.o.c.} \end{cases}$
3: $\{\mathcal{G}_1^*, \ldots, \mathcal{G}_r^*\} \leftarrow$ grafos vacíos definidos cobre $\mathcal{X}$
4: $\mathcal{G}_{tw \leq t}^* \leftarrow \emptyset$
5: **while** $\mathcal{E} \neq \emptyset$ **do**
6:     $a_{jk} \leftarrow \arg\max_{a_{jk} \in \mathcal{E}} \#(a_{jk})$
7:     $\mathcal{E} \leftarrow \mathcal{E} \setminus \{a_{jk}\}$
8:     $\mathcal{G}_i^* \leftarrow \begin{cases} \mathcal{G}_i^* \cup \{a_{jk}\}, \forall i : a_{jk} \in \text{enlaces}(\mathcal{G}_i) & \text{si } rep \\ \mathcal{G}_i^* \cup \{a_{jk}\}, \text{for } \exists! i : a_{jk} \in \text{enlaces}(\mathcal{G}_i) & \text{e.o.c.} \end{cases}$
9:     $\sigma \leftarrow$ un orden para $\mathcal{X}$      ▷ Usa [14]
10:     $\mathcal{G}_i^\sigma \leftarrow A(\mathcal{G}_i^*, \sigma) \ \forall i \in \{1, \ldots, r\}$
11:     **if** $\text{tw}(\bigcup_{i=1}^{r} \mathcal{G}_i^\sigma) \leq t$ **then**
12:         $\mathcal{G}_{tw \leq t}^* \leftarrow \bigcup_{i=1}^{r} \mathcal{G}_i^\sigma$
13:     **else**
14:         $\mathcal{G}_i^* \leftarrow \begin{cases} \mathcal{G}_i^* \setminus \{a_{jk}\}, \forall i : a_{jk} \in \text{enlaces}(\mathcal{G}_i) & \text{si } rep \\ \mathcal{G}_i^* \setminus \{a_{jk}\}, \text{for } \exists! i : a_{jk} \in \text{enlaces}(\mathcal{G}_i) & \text{e.o.c.} \end{cases}$
15:     **end if**
16: **end while**
17: Return $\{\mathcal{G}_1^*, \ldots, \mathcal{G}_r^*\}, \mathcal{G}_{tw \leq t}^*$

---

Debido al crecimiento exponencial del espacio de búsqueda con el tamaño de $|\mathcal{E}|$, el algoritmo utiliza un enfoque en el que se intenta maximizar los resultados obtenidos por iteración. Se evita usar

tamaños de población o iteraciones excesivas, priorizando una inicialización eficiente y operadores específicos para maximizar la efectividad.

Las características clave del algoritmo genético incluyen una inicialización híbrida, combinando soluciones heurísticas del algoritmo voraz (Alg. 2) con cromosomas aleatorios para mejorar la diversidad y la calidad; operadores específicos, diseñados para adaptarse a la naturaleza del espacio de búsqueda sin ser excesivamente restrictivos; y un *fitness* guiado por las restricciones, penalizando las soluciones que superan la restricción de *treewidth* para orientar la población hacia soluciones viables. A continuación, se detallan los operadores del algoritmo genético.

### 4.3.1 Inicialización

La inicialización de la población (tamaño $tamPob$) combina soluciones heurísticas del algoritmo *voraz de poda de arcos pre-fusión* (Alg. 2) con muestreo probabilístico para garantizar calidad y diversidad. El algoritmo voraz agrega iterativamente arcos respetando las restricciones de *treewidth*. Con él generamos dos cromosomas: uno para *treewidth* $t$ y otro para $t - 1$ (si $t > 2$).

Los cromosomas restantes se inicializan probabilísticamente según las frecuencias de los arcos $\#(a_{jk})$ en las redes de entrada[4]. Para cada arco $a_{jk} \in \mathcal{E}$, su probabilidad de inclusión es $\mathbb{P}(C(a_{jk}) = 1) = \frac{1}{1-\log(x)}$, donde $x$ es el valor normalizado min-max[5] de $\#(a_{jk})$.

### 4.3.2 Cruce

El cruce empareja a los individuos seleccionados y aplica un cruce de un solo punto aleatorio, dividiendo los cromosomas en dos partes y combinándolas para generar una nueva población con $tamPob$ individuos.

### 4.3.3 Mutación

En la etapa de mutación, todos los cromosomas son candidatos a modificaciones. Las probabilidades de agregar o eliminar arcos se ajustan dinámicamente según la relación $x = \frac{\text{tw}(\mathcal{G}_C^*)}{t}$, equilibrando exploración y explotación. La probabilidad de mutación se define de la siguiente forma:

$$\mathbb{P}(\text{añadir}) = \begin{cases} \left(\frac{x-1}{2}\right)^2 + 0.01, & \text{si } x \leq 1, \\ \frac{0.01}{x}, & \text{si } x > 1, \end{cases} \qquad \mathbb{P}(\text{eliminar}) = \begin{cases} \frac{x}{10}, & \text{si } x \leq 1, \\ \frac{\log(x)}{3} + 0.01, & \text{si } x > 1. \end{cases}$$

Cada arco se muta independientemente. Para agregar un arco, un arco candidato $a_{jk} \notin \mathcal{E}_C$ se agrega con probabilidad $\mathbb{P}(\text{añadir})$. Para eliminar un arco, un arco existente $a_{jk} \in \mathcal{E}_C$ se elimina con probabilidad $\mathbb{P}(\text{eliminar})$. Este enfoque adaptativo favorece la exploración equilibrada del espacio de búsqueda y dirige las soluciones hacia la región factible con *treewidth* $\leq t$.

### 4.3.4 Actualización de la población

Después de la mutación, la población se actualiza utilizando una estrategia elitista. Los dos individuos menos aptos son reemplazados por $C^*$, el mejor cromosoma encontrado durante el proceso evolutivo, y $C^l$, el mejor de la generación anterior. Este enfoque asegura que el material genético de alta calidad persista a lo largo de las generaciones, manteniendo diversidad para evitar convergencias prematuras.

## 5 Evaluación experimental

Nuestro enfoque experimental sigue el de [14, 15], ampliado con RBs más grandes y nuevas métricas.

### 5.1 Redes / conjuntos de datos

Siguiendo [15], evaluamos con redes sintéticas y reales. Los DAGs de entrada $\mathbb{G} = \{\mathcal{G}_1, \ldots, \mathcal{G}_r\}$ se generan a partir de un DAG base $\mathcal{G}_0$ aplicando el método de [14, 15]. Cada red de $\mathbb{G}$ sufre $p = n \cdot 0.75$ perturbaciones, donde $n$ es el número de nodos. En cada perturbación, se agrega o

---

[4] Por abuso de notación, $a_{jk}$ hace referencia tanto a $a_{jk}$ como a $a_{jk}^i$ en este párrafo.

[5] Cuando $\#(a_{jk})$ es igual para todos los arcos, se lleva a cabo una inicialización aleatoria uniforme ($\mathbb{P} = 0.5$).

elimina aleatoriamente un arco $\{X \to Y\}$, garantizando aciclicidad. La complejidad de la RB se limita a 3 padres, 4 hijos y $e = n \cdot 2.5$ arcos por nodo, manteniendo similitud estructural con $\mathcal{G}_0$. Para las redes sintéticas, $\mathcal{G}_0$ se genera con el método de [12], como en [14, 15]. Para las redes reales, utilizamos las siete RBs del *Bayesian Network Repository* de bnlearn[6] con entre 20 y 56 nodos[7].

## 5.2 Algoritmos

Este estudio evalúa el rendimiento de los algoritmos genético y voraz para el consenso de RBs bajo una restricción de *treewidth*, considerando varias configuraciones del conjunto de arcos candidatos:

- $\mathcal{E}_{\mathcal{G}^+}$**:** Los arcos provienen de la red fusionada $\mathcal{G}^+$ como en [15]. Los algoritmos de [15] se adaptan para optimizar métricas respecto a los grafos de entrada originales en lugar de $\mathcal{G}^+$.
- $\mathcal{E}_{\mathbb{G}}$**:** Los arcos son únicos entre las redes de entrada, $\mathcal{E}_{\mathbb{G}} = \bigcup_{i=1}^{r} \text{enlaces}(\mathcal{G}_i)$. Se aplican el algoritmo genético propuesto (Alg. 1) y su base voraz (Alg. 2) con decisiones de inclusión uniformes (rep = False).
- $\mathcal{E}_{\mathbb{G}}^*$**:** Los arcos incluyen repeticiones de cada red de entrada, $\mathcal{E}_{\mathbb{G}}^* = \bigsqcup_{i=1}^{r} \text{enlaces}(\mathcal{G}_i)$. Se aplican Alg. 1 y Alg. 2 para permitir decisiones de inclusión independientes (rep = True).

## 5.3 Reproducibilidad

Los algoritmos han sido implementados en Java (OpenJDK 17) con la biblioteca Tetrad 7.6.5[8]. El código, RBs y conjuntos de datos están disponibles en GitHub[9]. Los datos, además, también están disponibles en OpenML[10]. Los experimentos se han ejecutado en Rocky Linux 8.9 con procesadores AMD EPYC 7453, usando siete hilos y 8 GB de RAM por ejecución.

## 5.4 Metodología

Escalamos el trabajo de [15] evaluando RBs más grandes ($n = \{10, 30, 50\}$ en las sintéticas y desde 20 hasta 56 en las reales), y con un mayor número de DAGs de entrada ($r = \{10, 30, 50\}$). Evaluamos la fusión con dos métricas (Sección 2.3): SMHD y FSIM promedio respecto a los DAGs originales, denotadas como $m = \{\text{SMHD}_{\mathbb{G}}, \text{FSIM}_{\mathbb{G}}\}$. Para cada DAG base $\mathcal{G}_0$, generamos diez conjuntos de DAGs de entrada $\mathbb{G}^i = \{\mathcal{G}_1^i, \ldots, \mathcal{G}_r^i\}$ utilizando distintas semillas aleatorias. Para cada combinación $(i, r, n, m, tw)$, con $tw \in \{t \mid 2 \leq t < \text{tw}(\mathcal{G}^+)\}$, ejecutamos cada algoritmo genético y su correspondiente voraz descritos en la Sección 5.2. Los algoritmos genéticos emplean un tamaño de población de 100, en línea con la configuración óptima identificada en [15].

## 5.5 Resultados para RBs sintéticas

Se han realizado dos ejecuciones independientes de los algoritmos genéticos, una optimizando $\text{SMHD}_{\mathbb{G}}$ y otra $\text{FSIM}_{\mathbb{G}}$. Las Figs. 5 y 6 resumen los resultados para cada métrica, mostrando el desempeño promedio bajo diferentes restricciones de *treewidth*, así como el rango de *treewidths* $\text{tw}(\mathbb{G})_{min}^{max}$ y el *treewidth* promedio $\overline{\text{tw}(\mathbb{G})}$ de los grafos de entrada. La Tabla 2 proporciona un resumen detallado, incluyendo la diferencia media entre el algoritmo con mejor desempeño y los demás ($\overline{\text{DIF}}$), el número de veces que cada algoritmo alcanzó el mejor rendimiento, incluyendo empates (#MEJOR), y el tiempo promedio de ejecución ($\overline{\text{TIEMPO (S)}}$). Como referencia, el método de fusión sin restricciones $\mathcal{G}^+$ presenta una diferencia media de 211.5 en $\text{SMHD}_{\mathbb{G}}$ y 98.9 en $\text{FSIM}_{\mathbb{G}}$ respecto al mejor algoritmo en todos los conjuntos de datos. A partir de estos resultados, extraemos las siguientes conclusiones:

**Alg. 1 vs. [15]** El Algoritmo 1 sin repetición ($\mathcal{E}_{\mathbb{G}}$) obtiene los mejores resultados en todas las configuraciones, con gran estabilidad en $\text{SMHD}_{\mathbb{G}}$ y $\text{FSIM}_{\mathbb{G}}$, aunque con tiempos de ejecución algo mayores que el algoritmo genético de [15]. Por otro lado, el Algoritmo 1 con repetición de enlaces ($\mathcal{E}_{\mathbb{G}}^*$) sobresale en límites de *treewidth* más altos, pero presenta dificultades con los más pequeños.

---

[6]https://www.bnlearn.com/bnrepository/

[7]CHILD, INSURANCE, WATER, MILDEW, ALARM, BARLEY y HAILFINDER.

[8]https://github.com/cmu-phil/tetrad/releases/tag/v7.6.5

[9]https://github.com/ptorrijos99/BayesFL

[10]https://www.openml.org/search?type=data&tags.tag=bnlearn

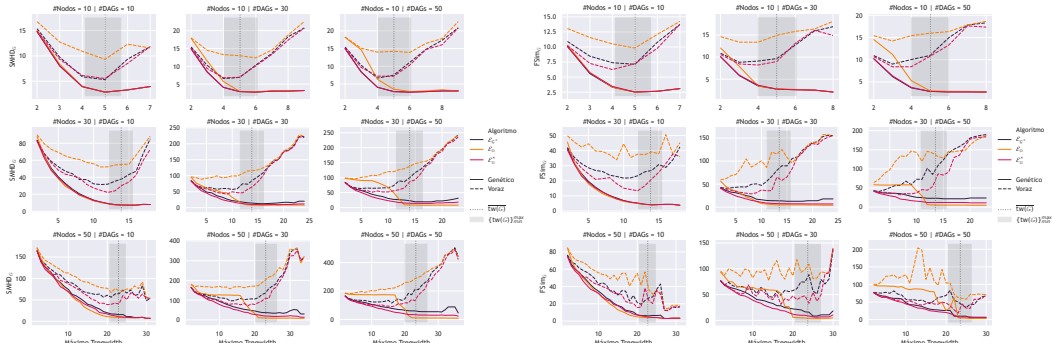

Figura 5: SMHD$_\mathbb{G}$ en RBs sintéticas.

Figura 6: FSIM$_\mathbb{G}$ en RBs sintéticas.

Tabla 2: Resultados en RBs sintéticas.

Tabla 3: Resultados en RBs reales.

| ALGORITMO | | SMHD$_\mathbb{G}$ | | | FSIM$_\mathbb{G}$ | | | SMHD$_\mathbb{G}$ | | | FSIM$_\mathbb{G}$ | | |
|---|---|---|---|---|---|---|---|---|---|---|---|---|---|
| | | DIF | #MEJOR | TIEMPO (S) | DIF | #MEJOR | TIEMPO (S) | DIF | #MEJOR | TIEMPO (S) | DIF | #MEJOR | TIEMPO (S) |
| VORAZ | $\mathcal{E}_{\mathcal{G}^+}$ [15] | 85.8 | 0 | **0.1** | 39.7 | 0 | **0.1** | 179.5 | 0 | **0.2** | 71.1 | 0 | **0.2** |
| | $\mathcal{E}_\mathbb{G}^*$ (Alg. 2) | 114.5 | 0 | 127.8 | 62.7 | 0 | 274.6 | 192.3 | 0 | 167.5 | 96.5 | 0 | 128.2 |
| | $\mathcal{E}_\mathbb{G}$ (Alg. 2) | 73.0 | 1 | 8.5 | 30.9 | 1 | 14.3 | 160.4 | 2 | 18.4 | 51.5 | 1 | 13.0 |
| GENÉT. | $\mathcal{E}_{\mathcal{G}^+}$ [15] | 12.6 | 27 | 4.1 | 5.4 | 30 | 1785.9 | 22.5 | 27 | 9.2 | 11.2 | 30 | 1703.0 |
| | $\mathcal{E}_\mathbb{G}^*$ (Alg. 1) | 15.1 | **95** | 250.0 | 9.5 | **91** | 2386.2 | **9.3** | **379** | 397.1 | 7.2 | **278** | 1970.5 |
| | $\mathcal{E}_\mathbb{G}$ (Alg. 1) | **3.5** | **95** | 50.5 | **1.4** | 90 | 1790.5 | 12.6 | 176 | 160.8 | **4.0** | 178 | 1596.1 |

**Algoritmos Voraces**   El Algoritmo voraz 2 con repetición de enlaces ($\mathcal{E}_\mathbb{G}^*$) tiene un rendimiento deficiente debido al mayor espacio de búsqueda, lo que afecta también al algoritmo genético correspondiente. Sin embargo, los algoritmos genéticos consiguen mitigar estas limitaciones.

**SMHD$_\mathbb{G}$ vs. FSIM$_\mathbb{G}$**   Los resultados para SMHD$_\mathbb{G}$ y FSIM$_\mathbb{G}$ están altamente correlacionados, con diferencias principalmente en los tiempos de ejecución. FSIM$_\mathbb{G}$ es más costoso computacionalmente debido a la necesidad de un número elevado de fusiones en cada iteración para evaluar cada *fitness*.

### 5.6   Resumen de resultados en RBs reales

La Tabla 3 resume la precisión estructural media y el tiempo de ejecución de los modelos generados para las RBs reales. Las métricas son las mismas que las usadas anteriormente en la Tabla 2. Las conclusiones son similares a las de las redes sintéticas. El Algoritmo 1 con repetición de enlaces ($\mathcal{E}_\mathbb{G}^*$) logra mejores soluciones al permitir mayor flexibilidad en la selección de arcos, pero a costa de un mayor tiempo de ejecución. Lidera en la diferencia media respecto a la mejor solución y tiene más victorias en SMHD$_\mathbb{G}$. Para FSIM$_\mathbb{G}$, aunque lidera en victorias, es superado ligeramente por el algoritmo sin repeticiones ($\mathcal{E}_\mathbb{G}$). Ambos algoritmos genéticos superan a [15], que, aunque más rápido en SMHD$_\mathbb{G}$, obtiene peores resultados en calidad de solución. La brecha entre algoritmos voraces y genéticos es aún mayor, destacando la robustez de los métodos propuestos.

## 6   Conclusiones

Este estudio propone un nuevo marco para la fusión de RBs, redefiniendo la tarea como alcanzar un consenso estructural bajo una restricción de treewidth. Los métodos tradicionales [14] tienden a generar redes excesivamente complejas, mientras que los enfoques restringidos [15] pueden sobreajustarse al ruido de entrada. Nuestro enfoque basado en consenso aborda estos problemas al centrarse en las estructuras compartidas y minimizar la divergencia entre la red fusionada y las RBs de entrada. Para resolver este problema, diseñamos dos algoritmos genéticos especializados con estrategias de inicialización, operadores genéticos específicos para el problema y una función de *fitness* que equilibra la similitud estructural con las restricciones de treewidth. Además, introdujimos métricas para evaluar la calidad del consenso mientras controlamos la complejidad de las RBs.

Los resultados experimentales en redes sintéticas y reales muestran que los algoritmos genéticos propuestos superan a los métodos previos [15], destacando los beneficios de podar enlaces antes de la fusión. El algoritmo que permite repeticiones de enlaces, aunque menos estable, a menudo obtiene los mejores resultados. En conclusión, los algoritmos genéticos propuestos son una solución robusta y flexible para el consenso restringido de RBs, con posibles mejoras en heurísticas voraces de inicialización, estrategias de optimización y aplicaciones en aprendizaje federado y otros dominios.

## Agradecimientos y Declaración de Financiación

Este trabajo está financiado por: TED2021-131291B-I00, PID2022-139293NB-C32 and FPU21/01074 (MICIU/AEI/10.13039/501100011033 and Unión Europea NextGenerationEU/PRTR), SBPLY/21/180225/000062 (Junta de Comunidades de Castilla-La Mancha y ERDF, UE) y 2022-GRIN-34437 (Universidad de Castilla-La Mancha y ERDF, UE).

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
