# OpenReview forum: "Algoritmos Genéticos para la Fusión Eficiente de Redes Bayesianas mediante Eliminación de Enlaces Previa a la Fusión"
_MAEB/2025/Congreso — MAEB 2025_

### Official Review · Reviewer_3UpF · 2025-03-14
**Algoritmos genéticos para resolver el problema de la fusión de redes bayesianas mediante restricciones sobre el treewidth.**

**Rating:** 4
**Confidence:** 4

**Review:**

El artículo estudia el uso de algoritmos genéticos para abordar el problema de fusión de redes bayesianas imponiendo restricciones en el treewidth. Comparan sus propuestas frente a propuestas heurísticas y discuten como la propuesta genética obtiene los mejores resultados. Cabe resaltar, que los autores aseguran la reproducibilidad compartiendo tanto el código como las instancias.

# Comentarios menores

- (4) Pondría en castellano en el abstract qué es treewidth.
- (24) Define IA como Inteligencia Artificial (IA).
- (61) Añadiría el enlace de nuevo en el footnote de la reproducibilidad, aunque esté puesto más tarde, facilita al lector.
- (67,68) Agradecería una descripción de la sección de preliminares para poder ver que me encontraré. Entiendo el problema de espacio que puede ocasionar, pero en caso de tenerlo creo que mejoraría la lectura.
- (94) "más óptimo" no es un término correcto, por favor, corrígelo.
- (203) Estructura del Algoritmo → Estructura del algoritmo.
- (216) Añade el punto y final.
- (243) Ampliaría si es posible un poco más.

# Comentarios

- ¿Existen modelos que puedan modelar el problema y mostrar el óptimo?
- (122) ¿Qué son grafos morales?
- (124) Hablas del uso del orden topológico, este algoritmo puede devolver diferentes soluciones válidas, ¿se contemplan todos los posibles ordenes topológicos?
- (251) ¿Tiene sentido evaluar con redes con mayor cantidad de nodos? ¿Cómo escala, por ejemplo, si se usan instancias de 10, 100, 1000?
- (293) El tiempo para comparar los algoritmos difiere, qué ocurre si se deja el voraz (si no fuese posible, aleatorizando con un algoritmo, por ejemplo GRASP entre otros) el mismo tiempo y se comparan. ¿Se mantienen los resultados?

---

### Official Review · Reviewer_14Uz · 2025-03-17
**Aproximación genética en fusión de redes bayesianas**

**Rating:** 4
**Confidence:** 4

**Review:**

El artículo presenta una propuesta para la fusión de redes bayesianas a través de un algoritmo genético. El problema es muy concreto y técnico. Desafortunadamente, las secciones 2 y 3 son demasiado complejas para un investigador que desconozca el ámbito de las redes bayesianas.

Respecto a los algoritmos presentados, se habla en el artículo de dos algoritmos genéticos. Sin embargo, solamente se presenta uno. Sí se presentan dos codificaciones diferentes, pero eso no justificaría la propuesta de dos algoritmos. Se agradecería una aclaración en este punto.

En cuanto al algoritmo voraz, se interpreta como una contribución del artículo, pero no queda claro su origen.

La figuras 5 y 6 son demasiado pequeñas y no se aprecian los valores de t usados en los experimentos a menos que se haga un zoom muy grande. Se sugiere ampliar tamaño de fuente en las imágenes. La descripción de las tablas resumen no explican cómo agrupan los diferentes valores de t.

Erratas:

- Pg. 3 "resultado más óptimo". Óptimo es un superlativo, nada puede ser más o menos óptimo. O es óptimo, o no lo es. Se sugiere utilizar "resultado mejor".

- Pg. 3: usar siempre el mismo estilo de mayúsculas/cursiva para "Método A"

-Pg, 6: falta un punto al final del último párrafo.

---

### Official Review · Reviewer_Nqp1 · 2025-03-17
**El trabajo aborda la fusión de Redes Bayesianas (RBs) mediante una metodología de consenso mediante la eliminación selectiva de enlaces en las RBs originales antes de la fusión. Para ello, se diseña un algoritmo genético con operadores específicos adaptados al problema. El trabajo está bien redactado y expone de manera clara cuáles sus contribuciones, destacando las principales diferencias con respecto a la literatura existente. Los resultados computacionales corroboran la eficacia del enfoque propuesto, demostrando un rendimiento superior en comparación con los métodos previos.**

**Rating:** 5
**Confidence:** 3

**Review:**

El trabajo aborda la fusión de Redes Bayesianas (RBs) mediante una metodología de consenso mediante la eliminación selectiva de enlaces en las RBs originales antes de la fusión. Para ello, se diseña un algoritmo genético con operadores específicos adaptados al problema. El trabajo está bien redactado y expone de manera clara cuáles sus contribuciones, destacando las principales diferencias con respecto a la literatura existente. Los resultados computacionales corroboran la eficacia del enfoque propuesto, demostrando un rendimiento superior en comparación con los métodos previos. Se propone la aceptación del trabajo.

Una cuestión menor a corregir es la frase "... logra un resultado más óptimo ..." de la línea 94. El término "óptimo" ya implica el máximo/mínimo grado de mejora posible. Por lo tanto, no es correcto decir más o menos óptimo.

---

### Decision · Program_Chairs · 2025-03-20

Accept